# Upregulation of Polyamine Transport in Human Colorectal Cancer Cells

**DOI:** 10.3390/biom10040499

**Published:** 2020-03-25

**Authors:** Misael Corral, Heather M. Wallace

**Affiliations:** School of Medicine, Medical Sciences and Nutrition, Institute of Medical Sciences, University of Aberdeen, Aberdeen AB25 2ZD, UK; corralmsc@gmail.com

**Keywords:** polyamines, putrescine, spermidine, uptake, DFMO, transport, drug delivery, colorectal cancer

## Abstract

Polyamines are essential growth factors that have a positive role in cancer cell growth. Their metabolic pathway and the diverse enzymes involved have been studied in depth in multiple organisms and cells. Polyamine transport also contributes to the intracellular polyamine content but this is less well-studied in mammalian cells. As the polyamine transporters could provide a means of selective drug delivery to cancer cells, a greater understanding of polyamine transport and its regulation is needed. In this study, transport of polyamines and polyamine content was measured and the effect of modulating each was determined in human colorectal cancer cells. The results provide evidence that upregulation of polyamine transport depends on polyamine depletion and on the rate of cell growth. Polyamine transport occurred in all colorectal cancer cell lines tested but to varying extents. The cell lines with the lowest basal uptake showed the greatest increase in response to polyamine depletion. Kinetic parameters for putrescine and spermidine suggest the existence of two separate transporters. Transport was shown to be a saturable but non-polarised process that can be regulated both positively and negatively. Using the polyamine transporter to deliver anticancer drugs more selectively is now a reality, and the ability to manipulate the polyamine transport process increases the possibility of using these transporters therapeutically.

## 1. Introduction

Polyamines are small molecules found in all eukaryotic cells and are important in several crucial biological processes ranging from nucleic acid stabilisation to cell proliferation [1,2,3,4].

Highly proliferating tissue and cells, as found in cancer, require a constant provision of polyamines to support their continuous proliferation. Many types of human cancer have been shown to have intracellular polyamine contents 4- to 6-fold greater than the corresponding normal tissue [3].

Polyamine homeostasis is complex. Intracellular concentrations are determined by a combination of *de novo* synthesis and transport of polyamines into and out the cell with each part being regulated carefully to maintain optimum cell growth and/or survival. Transport of nutrients, precursors and xenobiotics is an essential biological process and can be an active or passive process. Active transport is mediated by carrier proteins, which are present, to various extents, on the surface of cells. It requires energy and can be modulated depending on the needs of the cell. Passive transport is generally slower and can occur without carrier molecules via pores in the membrane [5].

Polyamines can either enter or exit the cell in accordance with the needs of the cell. Since polyamines have net positive charge at physiological pH, a transport system is required in order to take up exogenous polyamines and/or remove excess polyamines out of the cell [6]. While the reactions involved in the polyamine biosynthesis and catabolism have been described in depth, the mammalian polyamine transport system (PTS) remains less well-understood.

Polyamines have been shown to be closely related to cancer for many years now. Cancer patients exhibit elevated concentrations of polyamines in body fluids, especially in their acetylated form [7,8,9]. This relationship between cancer and polyamines has opened the door for polyamines as cancer biomarkers but more likely as markers of response rather than of diagnosis [10]. Cancer cells also have upregulated ornithine decarboxylase (ODC) [11], which confers a higher capacity for polyamine synthesis to cope with the demand for continuous proliferation.

Due to the link between polyamines and cancer cell growth, the polyamine metabolic pathway has been a target for anticancer strategies. One of the most iconic examples is that of α-difluoromethylornithine (DFMO). DFMO is a suicidal inhibitor of ornithine decarboxylase, the first and rate limiting step in polyamine biosynthesis. Despite DFMO showing great success as an anticancer therapy in vitro, it failed when tested in vivo. The principal reason for this is that DFMO triggers upregulation of the transport of exogenous polyamines that come from either the diet and/or the microbiome. This uptake thus counteracts the polyamine depletion caused by DFMO.

Although this was a disadvantage for DFMO as a monotherapy, it widens the possibility of using the polyamine transport as a means of delivering polyamine-conjugates or polyamine drug-like molecules to cells. In this study, the ability of the transport system to be regulated was investigated in order to better understand how this system could be used as a drug delivery system in the future.

## 2. Materials and Methods

### 2.1. Cell Culture

Human colorectal cancer cells (ECACC) were grown in Dulbecco’s modified Eagle’s medium (DMEM) or minimum essential medium Eagle (EMEM) supplemented with 10% (v/v) foetal bovine serum under standard conditions (37 °C, 5% CO_2_). Cells were routinely sub-cultured every 4 days with change of medium every 48 h and were seeded at 2.4 × 10^4^ cells/cm^2^ in 6-cm-diameter dishes for growth and polyamine content determination and in 24-well plates for uptake measurements.

### 2.2. Extraction of Polyamine and Proteins

Polyamine extraction was performed by resuspending the cell pellet in 0.2 M perchloric acid (PCA) and placing it on ice for 20 min to allow the extraction of acid-soluble content and protein precipitation to occur. After this time, tubes were centrifuged and the acid fraction was transferred to a clean reaction tube and stored at −20 °C until analysis. The remaining precipitate was dissolved in 0.3 M NaOH and incubated overnight at 37 °C and finally used for quantification of the total protein content.

### 2.3. Total Cellular Protein Determination

Total cellular protein content was determined by a modified method from Lowry [12] using a 96-well plate. A standard curve for protein was prepared in the range of 0 to 250 µg/ml from a 0.5 mg/ml stock of Bovine Serum Albumin (BSA) in 0.3 M NaOH. Samples were exposed to basic solution containing Cu^++^ for 15 min prior to adding 0.13 M Folin–Ciocalteau reagent and incubating in the dark for 30 minutes and then analysed using a Tecan Sunrise colorimetric spectrophotometric (Tecan Group Ltd, Männedorf, Switzerland) plate reader at 690 nm. The total protein content was expressed in mg/culture.

### 2.4. Quantification of Polyamines

Samples and standards were treated for dansylation as described by Li et al. [13] in a method developed in our laboratory. In total, 100 µM 1, 7-diaminoheptane was added as internal standard to each tube, plus 50 µl of 1 g/ml sodium carbonate and 500 µl of freshly prepared 10 mg/ml dansyl chloride in acetone. Tubes were left overnight at 37 °C. On the next day, 0.5 ml of toluene was added to extract the dansylated products. Organic phases were evaporated to dryness under a N_2_ flow, reconstituted in 200 µl methanol and vortex-mixed for 10 seconds. LC-MS (Thermo Scientific, Hemel Hempstead, UK) analysis was performed and results were expressed in nmol/mg of protein.

### 2.5. Polyamine Uptake

Polyamine uptake was started by the addition of radiolabelled [^3^H] polyamines at different time points with a final concentration of 5.55 kBq/well. Cells were harvested, and the content of the well was transferred to a reaction tube and the wells were rinsed with 500 µl of ice-cold phosphate buffered saline (PBS), which was added to the tubes as well.

All the samples were centrifuged at 3,500 g_av_ for 5 min and the supernatant was discarded. Cell pellets were rinsed with 500 µl of ice-cold PBS and centrifuged again. Supernatant was discarded and the pellets were resuspended in 300 µl of 0.2 M PCA and placed on ice for 20 minutes. After this time, tubes were centrifuged at 15,000 g_av_ for 5 min and the acid-soluble fraction was completely transferred to a clean reaction tube. The remaining pellet was dissolved in 300 µl of 0.3 M NaOH and the tubes were left at 37 °C overnight before determination of total protein content.

Acid fractions in 50-µl aliquots were transferred to scintillation tubes containing 2 ml of scintillation cocktail liquid and analysed. The specific activity of each radiolabelled polyamine was used to convert dpm to pmol, and the results were expressed in pmol of polyamine/mg of protein.

### 2.6. Transport Studies

In order to study the polarisation of transport, transwells were used. Cells were seeded in 12-well polycarbonate inserts at 10 × 10^4^ cell/cm^2^ with 0.5 ml of medium in the insert and 1.5 ml in the well. Medium was changed in the outer chamber every 48 h and transepithelial electrical resistance (TEER) was monitored with a Millicell® ERS-2 Voltohmmeter (Millipore Corporation, Billerica, MA, USA). The cell monolayer was considered complete when the reading of TEER was 750–850 Ω for at least two measurements at different days. After this time (14–16 days), radiolabelled polyamine was added to either the inner or outer chamber and then monitored at regular intervals by taking 5 µl from each chamber to a scintillation vial with 2.5 ml of scintillation fluid. At the end of the time course, cells were carefully scraped and all the content of the insert was transferred to a reaction tube and the inserts were rinsed with 500 µl of ice-cold PBS, which was added to the tubes.

All the samples were processed and analysed as per polyamine uptake protocol described above. Polycarbonate membranes from the inserts were cut and placed into scintillation vials as well to verify for non-specific binding.

### 2.7. Statistical Methods

Result values were shown as the mean of all replicate values ± standard error of the mean (SEM) in which the number of independent experiments was equal to or more than 3.

Statistical analysis was performed using GraphPad Software Prism version 8 (GraphPad, San Diego, CA, USA). Results were analysed by one-way analysis of variance (ANOVA) with Dunett’s post-tests. A p value less than 0.05 was considered as statistically significant.

## 3. Results

It is known that treatment with DFMO can upregulate polyamine transport in mammalian cells but the time needed to achieve this and the temporal relationship to polyamine depletion is not clear. In SW480 human colorectal cancer cells, the uptake of putrescine and spermidine was measured in exponentially growing cells and was shown to be time-dependent. The uptake of spermidine was much greater (11-fold) than that of putrescine (Figure 1: Control). In order to investigate the time needed for this increase, we exposed cells for 3–24 h to DFMO before measuring uptake (Figure 1). Previous studies have focused on longer-term (24 h plus) exposure to DFMO to investigate the effect on uptake. In the case of putrescine, as little as 3 h exposure resulted in an increase in uptake, while for spermidine, 18 h treatment was required before the increase was observed (Figure 1). For both putrescine and spermidine, polyamine uptake was saturated by 3–4 h. The maximum increase of uptake was approximately 6-fold for putrescine and 2-fold for spermidine (Figure 1).

The hypothesis is that increased uptake occurs in response to decreased intracellular polyamine content. This was tested using the timed exposure experiments where intracellular concentrations of each polyamine were measured at each time. Untreated cells showed little changes in their total polyamine content (Table 1) over 24 h, while DFMO-treated cells showed a time-dependent decrease in total polyamine content losing approximately 40% of their total polyamine content in 24 h (Table 1). Analysis of individual polyamine content showed that putrescine was depleted quickly being below the limit of detection by 3 h, whereas spermidine decreased more slowly reaching the limit of detection by 24 h (Figure 2). The decrease in the individual polyamine concentrations in the cells shows a clear alignment with the increase in uptake.

SW480 are human colorectal adenocarcinoma cells, and it was important to determine if these were typical of cells in terms of polyamine uptake. Four other human cancer cell lines and one normal cell line were tested for polyamine uptake and for their response to DFMO (Table 2). A range of uptake values were noted with Caco-2 having the greatest basal uptake and DLD-1 cells the least at 13.9 and 0.4 pmol/mg protein, respectively. DFMO increased the polyamine uptake in all cell lines with the cell line with the lowest basal uptake in untreated cells exhibiting the greatest increase in response to DFMO (Table 2).

Kinetic analysis using Michaelis–Menten methodology showed that both putrescine and spermidine exhibited a two-component uptake system: a classic saturable one and another non-saturable component that depended only on the concentration of the substrate. Kinetic parameters for putrescine and spermidine are shown in Table 3.

In both cases, the induced uptake showed a higher Vmax compared to the control. However, the affinity for putrescine was marginally higher in the induced uptake, but the affinity for spermidine was similar for both normal and induced uptake.

A number of inhibitors of polyamine uptake have been synthesised and two of these were investigated in this study, AMXT 1505 and 2030 (Figure 3). These novel competitive inhibitors of uptake were both effective in preventing both basal and DFMO-increased uptake. The extent of inhibition was shown to be slightly higher for the increased uptake in all cases (Table 4).

The transwell system provides a means to examine the polarity of uptake. For these studies, Caco-2 cells were used as these cells form tight junctions and non-permeable monolayers. The integrity of the monolayers was determined by measuring their transepithelial electrical resistance (TEER), which was maximal at 14–16 days and had a value of 750–850 ohms.

Putrescine uptake occurred from both sides of the cell monolayer. Apical uptake showed a linear trend while the basolateral presented a saturable pattern. When the polyamine uptake transport inhibitor, AMXT 2030, was added along with the radiolabel, uptake was reduced from both apical and basolateral sides almost completely (Figure 4a,b). Spermidine uptake exhibited similar patterns with uptake on both sides, although uptake on the apical side was much faster than that on the basolateral side. AMXT 2030 again inhibited uptake but to a lesser extent than for putrescine (Figure 4c,d)).

In order to determine the relationship among intracellular polyamine content, the amount of AZ protein present and the degree of uptake, SW480 cells were grown for varying lengths of time before being treated with DFMO for 24 h. The aim was to determine the effect of growth status (exponential or high density) on polyamine uptake.

In exponentially growing cells, uptake decreased gradually from 3.4 to around 0.5 pmol/mg protein at late log growth (96–120 h). DFMO-treated cells decreased from 12 pmol/mg protein until reaching similar uptake values as control at 144 and 168 h (Figure 5). This indicates that uptake is linked to cell growth and when growth is low (late times in culture) so is uptake regardless of DFMO stimulation.

Total polyamine content showed a similar behaviour in control cells with a decrease up to 96 h and then a plateau, while polyamine content of DFMO-treated cells remained consistent throughout the duration of the experiment but below the values of control cells.

## 4. Discussion

The aim of this study was to understand better the regulation of the polyamine transport system in colorectal cancer cells with the ultimate aim to harness the transporter as potential drug delivery system for novel anticancer agents.

It has been known for several years that DFMO enhances the uptake of polyamines [15,16], but this effect and its proportionality was never fully characterised. Although it was previously suggested that prior to stimulation of uptake, loss of polyamines had to occur, no study had demonstrated this clearly [17]. Our interest is in understanding the regulation of polyamine transport in order to use it therapeutically, thus it was important to determine how DFMO affected the system in greater detail. This study has shown that polyamine depletion and the increase of uptake occur in parallel.

The nature of this link appears to be dependent on the polyamines, as putrescine uptake was enhanced in a time-dependent manner from 3h onwards—the time sufficient to deplete all putrescine from the cell. Similarly, spermidine uptake only increased when spermidine was depleted, so after approximately 18 h. It is interesting that despite the decrease of total polyamine content by 33% after 12 h of DFMO exposure, this was not enough to enhance spermidine uptake. Rather a significant decrease (74%) specifically in the spermidine pool was required to increase its uptake.

Intracellular spermine content was maintained even when the cells were treated with DFMO, an observation that has been made numerous times before. This emphasises the importance of spermine for cell survival and it supports spermine interacting with nucleic acids as stabiliser [18]. An interesting question then is, “Can DFMO induce an increase in spermine uptake when this polyamine’s intracellular pool remained unchanged?” Unfortunately, it was not possible to investigate spermine uptake in this study.

From our determination of the physical constants of the transporter(s), it appears to be that the observed enhancement in uptake is due to increased velocity (V_max_) in the transport process for both polyamines tested. While the difference in K_m_ between putrescine and spermidine can exist in a single transporter, the differences in normal/induced uptake indicate different characteristics of the carrier transporting each of the polyamines, suggesting different carriers. The existence of at least two transporters has been proposed by other groups as well [19,20,21]

There is a debate whether the inward and outward transport of polyamines are mediated by similar or different carriers. Wallace et al. suggest the existence of separate transporters for uptake and export of polyamines after showing effects on the former but not on the later using competitive polyamine-uptake inhibitors on human cancer cells [22]. On the other hand, Sakata’s et al. conclusions on antizyme (AZ) regulation of the transporter(s) favour the utilisation of the same carrier for uptake and export [23,24].

Trying to contribute to this debate, we also investigated the sidedness of polyamine transport. Uptake showed little preference for apical or basolateral membranes occurring on both sides with, perhaps, a slight favouring of the apical side for spermidine uptake. The AMXT agents were designed to inhibit spermidine transport; however, in our model they were effective in inhibiting uptake of both putrescine and spermidine with a greater effect on putrescine uptake. As AMXT 2030 did not show significant inhibition in any of the cases when export of polyamines was observed (data not shown), this may suggest that the uptake and export processes are catalysed by different carriers.

With regard to the link between uptake and intracellular polyamine content, although it was true that for an increased uptake a decrease in polyamine content was necessary (as observed in previous experiments), at low growth rates (late times in culture) there was a decreased polyamine content but no increase in uptake. This was the case even in the presence of DFMO suggesting growth rate also regulates polyamine uptake.

Thus, in general, the trigger for an increased uptake of polyamines is a decrease in content, but the extent of this increase will be ruled by the growth status of the cell population, as well as the degree of polyamine depletion.

## 5. Conclusions

Polyamine transport occurs in all human colorectal cancer cells tested and the rate of uptake is enhanced in response to polyamine depletion. Uptake and depletion are temporally linked; however, growth status is a key regulator of uptake. Polyamine transport across different cell lines showed a pattern where the lowest uptake in the basal state corresponded to the greatest increase when uptake is upregulated via polyamine depletion but with an apparent maximum uptake.

Upregulation of the uptake via depletion of polyamines increased affinity of the transporter for putrescine, but not for spermidine, which instead increased its velocity. There was an order of magnitude of difference in affinity for the transporter between putrescine and spermidine. Transport was inhibitable but did not show polarity with putrescine and spermidine being taken up on both apical and basolateral membranes.

## Figures and Tables

**Figure 1 biomolecules-10-00499-f001:**
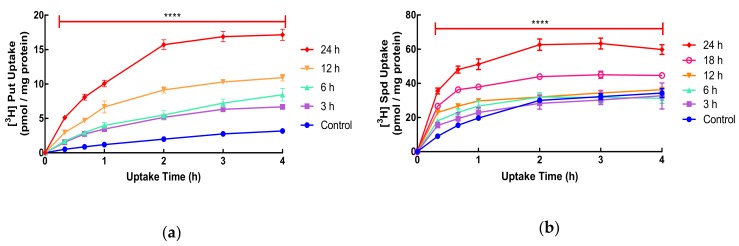
Uptake of putrescine **(a)** and spermidine **(b)** by SW480 cells. Cells were seeded at 2.4 × 10^4^ cell/cm^2^ on 24-well plates and grown for 48 h. Cells were pre-treated with DFMO (5 mM) for varying lengths of time (0–24 h). To measure uptake after exposure to DFMO, cells were incubated with radiolabelled polyamine at each DFMO exposure time for up to 4 h. The final concentration of radioactivity was 5.55 kBq/well (2.4 nM for putrescine or 5.6 nM for spermidine). Uptake was measured by liquid scintillation spectrometry. Values are mean ± SEM when *n* ≥ 3 or range when *n* < 3. Putrescine: (control and 24 h *n* = 6; 6 and 12 h *n* = 2; 3 h *n* = 1) with four replicates per experiment. Spermidine: (control and 24 h *n* = 3; 3 and 6 h *n* = 2; 12 and 18 h *n* = 1) with two replicates per experiment. For 24 h, *P*: ****<0.0001, compared to the respective control.

**Figure 2 biomolecules-10-00499-f002:**
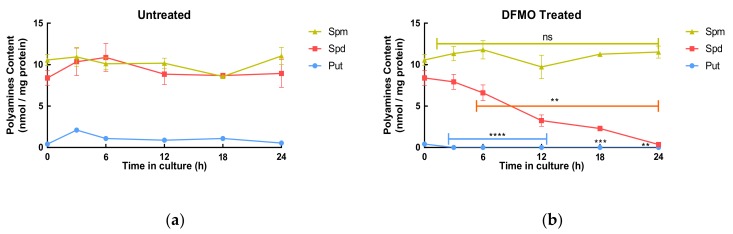
Individual polyamine content in untreated **(a)** and DFMO-treated cells **(b)**. SW480 cells were seeded at 2.4 × 10^4^ cell/cm^2^ in duplicate on 6-cm-diameter dishes and grown for 48 h. Where indicated, cells were pre-treated with DFMO (5 mM) for 24 h. Subsequently, two dishes were harvested and this time was set as t = 0 h. All plates were harvested at the time indicated and polyamine content was determined by LC-MS. Values are mean ± SEM (*n* = 3) with duplicates for each experiment. *P*: *<0.05, **<0.01, ***<0.001, ****<0.0001, ns = not significant; compared to the respective untreated control.

**Figure 3 biomolecules-10-00499-f003:**
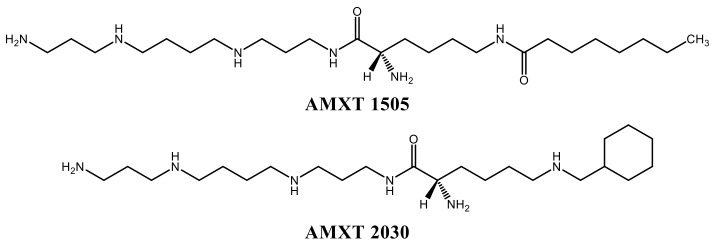
Molecular structure of the polyamine transport inhibitors AMXT 1505 and 2030 [14].

**Figure 4 biomolecules-10-00499-f004:**
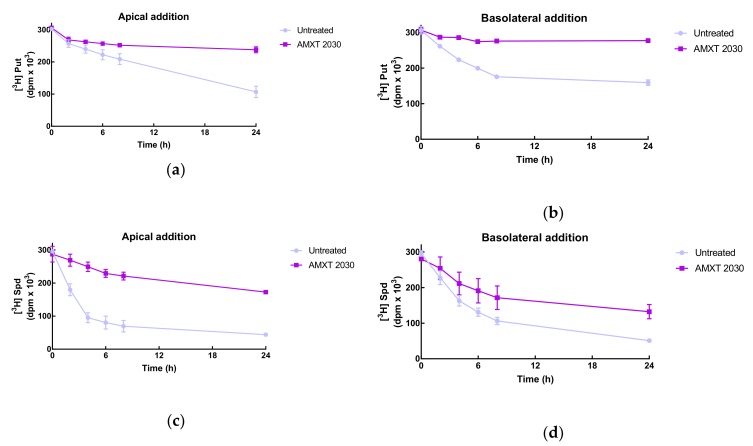
Putrescine **(a,b)** and spermidine **(c,d)** uptake from both sides of a Caco-2 cell monolayer. Cells were seeded at 10.0 × 10^4^ cell/cm^2^ in polycarbonate membrane transwell inserts in 12-well plates and grown for 14–16 days until the transepithelial electrical resistance (TEER) reading was consistently between 750–850 ohms. At this time, radiolabelled polyamine at a final radioactivity of 5.00 kBq/well (2.2 nM for putrescine or 5.0 nM for spermidine) was added with or without AMXT 2030 (10:1 ratio, 10.13 µM for putrescine, 23.5 µM for spermidine) into either the apical or basolateral chamber. This time was set as t = 0 h. Medium in 5 µl aliquots from each chamber were taken to measure total radioactivity by liquid scintillation spectrometry at regular intervals. Values are mean ± SEM when *n* ≥ 3 or range when *n* < 3. (Controls *n* = 3, AMXT *n* = 2).

**Figure 5 biomolecules-10-00499-f005:**
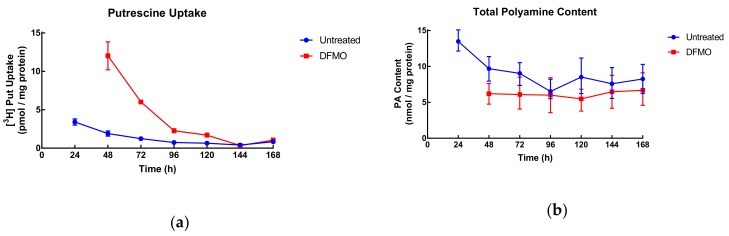
Uptake of putrescine **(a)** and total intracellular polyamine content **(b)** of SW480 cells. Cells were seeded at 2.4 × 10^4^ cell/cm^2^ on 24-well plates for uptake or 6-cm dishes for polyamine content analysis and grown for different times. Where indicated, cells were pre-treated with DFMO (5 mM) for 24 h. After the indicated growth time, 6-cm dishes were harvested for polyamine content analysis by LC-MS, and radiolabelled putrescine (2.4 nM) with a final radioactivity of 5.55 kBq/well was added to the 24-well plates and cells were incubated for one hour. Uptake was measured by liquid scintillation spectrometry Values are mean with range (*n* = 2) with duplicates for each experiment.

**Table 1 biomolecules-10-00499-t001:** The effect of DFMO on total polyamine content.

Time (h)	Untreated	DFMO	
	Total Polyamine Content (nmol/mg protein)		
0	19.40 ± 1.38	19.40 ± 1.38	ns
3	23.38 ± 2.66	19.26 ± 1.70	ns
6	22.05 ± 2.25	18.39 ± 2.02	ns
12	19.89 ± 1.53	12.98 ± 2.05	*
24	20.53 ± 2.55	11.88 ± 0.67	**

SW480 cells were seeded at 2.4 × 10^4^ cell/cm^2^ in duplicate on 6-cm-diameter dishes and grown for 48 h. Where indicated, cells were pre-treated with DFMO (5 mM) for 24 h. Subsequently, two dishes were harvested and this time was set as t = 0 h. All plates were harvested at the time indicated and polyamine content was determined by LC-MS. Values are mean ± SEM (*n* = 3) with duplicates for each experiment. *P*: *<0.05, **<0.01, ns = not significant, compared to the respective untreated controls. (One measurement was performed at 18 h with values of 18.33 and 13.56 nmol/mg protein for untreated and DFMO, respectively.)

**Table 2 biomolecules-10-00499-t002:** The effect of inhibition of PA biosynthesis on putrescine uptake in a range of colorectal cancer and normal cells.

Cell Line	Untreated	DFMO	Fold Increase
	Putrescine Uptake (pmol/mg protein)		
DLD-1	0.35 ± 0.01	14.09 ± 0.40	40.2
WiDr	2.45 ± 0.10	21.30 ± 0.42	8.7
SW480	3.47 ± 0.13	18.44 ± 0.94	5.3
CCD841CoN	6.09 ± 0.50	23.98 ± 2.89	3.9
HCT-116	8.68 ± 0.35	26.96 ± 3.08	3.1
Caco-2	13.88 ± 0.63	20.49 ± 0.74	1.5

CCD841CoN are normal human colonocytes. All other cell lines are human colorectal cancer cells. Cells were seeded at 2.4 × 10^4^ cell/cm^2^ on 24-well plates and grown for 48 h. Where indicated, cells were pre-treated with DFMO (5 mM) for 24 h. Subsequently, cells were incubated with radiolabelled putrescine (2.4 nM) for 4 h at a final concentration of radioactivity of 5.55 kBq/well. Uptake was measured by liquid scintillation spectrometry. Values are mean ± SEM when *n* ≥ 3 or range when *n* < 3. (*n* = 3 for SW480 and Caco-2; n=2 for the remaining) with two replicates per experiment.

**Table 3 biomolecules-10-00499-t003:** Kinetic parameters of the uptake transporter.

	Putrescine			Spermidine		
	Control	DFMO		Control	DFMO	
Km (µM)	3.68 ± 0.49	1.92 ± 0.11	*	0.35 ± 0.04	0.32 ± 0.03	ns
Vmax (nmol/h/mg protein)	2.60 ± 0.10	9.93 ± 0.14	****	2.12 ± 0.05	5.88 ± 0.10	****

SW480 cells were seeded at 2.4 × 10^4^ cell/cm^2^ on 24-well plates and grown for 48 h. Where indicated, cells were pre-treated with DFMO (5 mM) for 24 h. After the growth time, different concentrations of radiolabelled substrate with a final radioactivity of 5.55 kBq/well were added and incubated for 30 min. Uptake was measured by liquid scintillation spectrometry. All results were plotted using Michaelis–Menten analyses and the values shown were generated by regression analysis using Graphpad Prism 7 software. The values are mean ± SEM (*n* = 4) with two replicates per experiment. *P*: *<0.05, ****<0.0001, ns = not significant; compared to the respective controls.

**Table 4 biomolecules-10-00499-t004:** The effect of AMXT compounds on uptake.

AMXT	Putrescine		Spermidine	
	Control	DFMO	Control	DFMO
	Inhibition of Uptake (% Relative to Untreated Value)			
**1505**				
10 µM	59.7 (19.4)	78.4 (6.4)	27.3 ± 4.6	50.7 ± 1.4
25 µM	76.2 (2.6)	85.1 (6.6)	44.9 ± 2.5	60.9 ± 1.3
**2030**				
10 µM	68.8 (16.3)	82.7 (5.6)	39.6 ± 2.8	55.1 ± 1.7
25 µM	80.6 (9.2)	87.9 (7.1)	54.9 ± 2.5	63.3 ± 1.8

SW480 cells were seeded at 2.4 × 10^4^ cell/cm^2^ on 24-well plates and grown for 48 h. Where indicated, cells were pre-treated with DFMO (5 mM) for 24 h. After the growth time, radiolabelled putrescine or spermidine (10 µM) with a final radioactivity of 5.55 kBq/well and different concentrations of the inhibitors were added and incubated for one hour. Uptake was measured by liquid scintillation spectrometry. Results were expressed as % inhibition compared to untreated samples. Values are mean with range in brackets (*n* = 2) for putrescine and mean ± SEM (*n* = 3) for spermidine with two replicates per experiment.

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
