# Peer review of "Upregulation of Polyamine Transport in Human Colorectal Cancer Cells"

_biomolecules, 2020, doi:10.3390/biom10040499_

Round 1
Reviewer 1 Report
Corral et al. present a series of studies that explore the properties of polyamine transport, specifically putrescine and spermidine, in the context of varying levels of polyamine depletion and cell density. The impetus for this work is to develop a more complete understanding of polyamine transport, including the stimuli that trigger altered transport rates, to enable future exploitation of this system for delivery of therapeutic agents to malignant cells that exhibit robust polyamine uptake. The presented studies are well designed and informative. However, there are some weaknesses and a number of revisions and additions are necessary before this work is acceptable for publication.
Major comments
- The studies that include ornithine decarboxylase antizyme (AZ) amount to a single western blot in a single experiment and provide limited information. The changes in AZ levels are very minimal and weekly associate with polyamine uptake. As the authors state, cellular growth rate is the most likely driver of polyamine uptake and these experiments provide little evidence that this process includes AZ. What are the levels of ornithine decarboxylase in these cells over this time course, since AZ in complex with ODC may not be free to regulate transport? How are AZ levels changing upon DFMO treatment in other studies (Figure 1, 2) and in other cell lines (Table 2)? In addition, the Introduction does not contain any background on AZ or its regulation of PA transport (e.g. lines 292-295). This line of investigation should be supplemented with additional data or removed.
- Experiments with polyamine transport inhibitors are very difficult to follow. What are the initial publications and previous results acquired with these compounds? What are the units reported in this table? How are putrescine data presented (mean and range?)?
- There are no studies to probe the mechanism of increased polyamine uptake upon polyamine depletion. The authors should at least present some proposed molecular mechanisms, and any supporting or refuting data, within the Discussion. Is this associated with recruitment of existing proteins to the membrane, new protein synthesis, increased transcription of transport complex proteins, etc.? What are the sensors that detect polyamine depletion and trigger these molecular events?
- Line 277, what is the basis for the statement that the putrescine carrier is “less specific?” Figure 1 appears to show very selective increases in putrescine but not spermidine uptake at early time points after DFMO treatment.
Minor comments
- Line 54, consider converting “cells” to “cell”.
- Limited details were provided regarding the sources of chemicals, antibodies and cell lines used for these studies.
- Line 73, add was “dissolved” in 0.3 M NaOH.
- Line 102, what is the calculation to determine pmol of polyamine/mg of protein?
- Line 106, define TEER.
- A more descriptive title is needed for Figure 1. Both putrescine and spermidine incubation should be listed in the legend. What are the “different times” as referred to in this legend as well as Table 2 legend?
- Statistically significant p values are misstated throughout as greater than (p>0.05) rather than less than (p<0.05).
- Table 2 should be annotated to include cancer type of origin for all cell lines.
- Table 3, what are the methods, calculations and data used to derive the Km and Vmax values?
- Figure 3 legends are cut off in A and C. Limited data and explanations are provided to support the statements of “linear” vs “saturable” kinetics, especially given the lack of values between 8 and 24 hours.
- Figure 4, which cell line was utilized?
Author Response
Response to Reviewers
Thank you to all the reviewers for their careful reading of the manuscript and helpful comments. We have addressed these as indicated below.
Reviewer 1
Point 1
We are aware that the AZ data are not complete but we thought it added some interest. As we do not have the capability or time to carry out further experiments, we have removed this part from the manuscript.
Point 2
We have added further detail regarding structure and source of the AMXT compounds. The units were included in the table but we have enhanced this to make it clearer. Yes, putrescine is the mean and range.
Point 3
Our study was to look in more detail at the transport process. While the reviewer is correct that we could speculate on possibilities we did not think this was appropriate here as we had not developed the study in this direction.
Point 4
As this statement is unclear and potentially confusing, we have removed it from the text.
Minor comments
With the exception of number one all these have been corrected. We could not find the cells referred to in line 54.
Reviewer 2 Report
This study further investigates the phenomenon of DFMO-induced polyamine depletion and the subsequent induction of polyamine uptake. It is important to better understand the regulation of the polyamine transport system since it can be harnessed as a potential drug delivery system in polyamine-rich tumors. There are some concerns that should be addressed before publication:
- How was the DFMO concentration determined? Could this high concentration of DFMO lead to other effects besides lowering polyamine levels? Would lower concentrations of DFMO be sufficient to lower putrescine and spermidine levels, and is this cell type-dependent?
- The concentration of radiolabeled putrescine and spermidine should be stated for every experiment.
- Line 190: Please clarify the statement that “putrescine and spermidine exhibit a two-component uptake system – a classical saturable one and another non-saturable component that depended only on the concentration of the substrate.”
- Information concerning the polyamine transport inhibitors, AMXT 1501 and AMXT 2030 should be given, including structures and references. How do these 2 inhibitors differ?
- In Table 4, it appears that both AMXT 1501 and AMXT 2030 inhibit the DFMO-induced spermidine uptake more than the basal spermidine uptake. The sentence at line 205 should be revised. Also, clarify whether the range is given in parentheses in Table 4.
- The claim that uptake is linked to cell growth (line 236) is not adequately substantiated by the data. It is possible that polyamine uptake is decreased due to physical obstruction of the uptake system due to confluency of packed-in cells. The authors should used synchronized cells arrested at different stages of the cell cycle to say that polyamine uptake is governed by growth rate.
- What is the effect of either AMXT 1501 and AMXT 2030 on the natural inhibitor of polyamine transport – antizyme and its inhibitor, antizyme inhibitor? Please explain why antizyme is increased in cells grown 144 hr and not at 168 hr.
- In Materials and Methods, the names and sources of cell lines used in this study should be included. Also, include the source of antizyme antibody.
- Minor concerns: Sentence beginning at line 202 should be 2 sentences.
The word “plateau” is not spelled correctly in line 238.
Author Response
Reviewer 2
Point 1
The DFMO concentration was not measured in these studies. 5 mM DFMO is standard concentration used as this molecule is not actively taken up by cells rather it enters the cell by diffusion hence the high concentration required. Lower concentrations of DFMO can decrease putrescine and spermidine but the effects takes longer.
Point 2
The concentration is now stated for all experiments.
Point 3
The statement has been expanded and clarified to include Michaelis Menten analysis.
Point 4
The structures are now added together with references
Point 5
Table 4 has been clarified
Point 6
The point is well taken but in our experiments, confluence was not reached until the later times and so we believe that the cell growth rate and polyamine uptake were decreasing in parallel before confluence became an issue.
Point 7
As our AZ data is preliminary, as indicated by reviewer 1, we have removed this from the manuscript.
Points 8 and 9
Changes have been made as per the reviewer’s request
Reviewer 3 Report
The manuscript entitled “Upregulation of polyamine transport in human colorectal cancer cells” indicate that the lowest basal uptake showed the greatest increase in response to polyamine depletion, and that the different characteristics of the putrescine and spermidine transport suggest different carriers.
The results are interesting, but some modifications may be necessary.
- Results section: it may be better to add subheading.
- Since the effective concentrations of putrescine, spermidine and spermine are different, it may be better to change total polyamine content into putrescine, spermidine and spermine contents (Table 1 & 2, and Fig. 4).
- 6, l. 4. It may be better to add Table 4 in the sentence: ….. was increased or not. → ….. was increased or not (Table 4).
- Please add the reference or the structures of AMXT 1505 and 2030.
- Explanation of TEER is necessary: transepithelial electrical resistance.
Author Response
Reviewer 3
Point 1
After review, we think that subheadings would not be appropriate in this paper as the results section covers transport in general.
Point 2
We have clarified that the individual polyamine content in Figure 2 is derived from the same experiment as shown in Table 1.
Table 2 is the uptake of putrescine and not polyamine content as measured by LC-MS
Points 3, 4 and 5
Changes have been made as per the reviewer’s suggestion.